# Real-Time PCR Method as Diagnostic Tool for Detection of Periodontal Pathogens in Patients with Periodontitis

**DOI:** 10.3390/ijms25105097

**Published:** 2024-05-07

**Authors:** Sendi Kuret, Nina Kalajzic, Matija Ruzdjak, Blaženka Grahovac, Marina Adriana Jezina Buselic, Sanda Sardelić, Anja Delic, Lana Susak, Davorka Sutlovic

**Affiliations:** 1Department of Health Studies, University of Split, 21000 Split, Croatia; nkalajzic@ozs.unist.hr; 2Department of Biology, Faculty of Natural Science, University of Zagreb, 10000 Zagreb, Croatia; mruzdak@gmail.com; 3Medical Faculty, University of Rijeka, 51000 Rijeka, Croatia; blazenka.grahovac@gmail.com; 4Dental Practice Marina Adriana Jezina Buselic, 21000 Split, Croatia; marinajezina@gmail.com; 5Department of Microbiology, University Hospital Centre Split, 21000 Split, Croatia; sandasard@gmail.com; 6Polyclinic Analysis, 21000 Split, Croatia; anja0310@hotmail.com (A.D.); lana.susak.17@gmail.com (L.S.); 7Department of Toxicology and Pharmacogenetics, School of Medicine, University of Split, 21000 Split, Croatia

**Keywords:** periodontal disease, periodontopathogenic bacteria, DNA-strip technology, real-time PCR

## Abstract

The most common type of periodontal disease is chronic periodontitis, an inflammatory condition caused by pathogenic bacteria in subgingival plaque. The aim of our study was the development of a real-time PCR test as a diagnostic tool for the detection and differentiation of five periodontopathogenic bacteria, *Aggregatibacter actinomycetemcomitans*, *Porphyromonas gingivalis*, *Tannerella forsythia*, *Prevotella intermedia*, and *Treponema denticola*, in patients with periodontitis. We compared the results of our in-house method with the micro-IDent^®^ semiquantitative commercially available test based on the PCR hybridization method. DNA was isolated from subgingival plaque samples taken from 50 patients and then analyzed by both methods. Comparing the results of the two methods, they show a specificity of 100% for all bacteria. The sensitivity for *A. actinomycetemcomitans* was 97.5%, for *P. gingivalis* 96.88%, and for *P. intermedia* 95.24%. The sensitivity for *Tannerella forsythia* and *T. denticola* was 100%. The Spearman correlation factor of two different measurements was 0.976 for *A. actinomycetemcomitans*, 0.967 for *P. gingivalis*, 0.949 for *P. intermedia*, 0.966 for *Tannerella forsythia*, and 0.917 for *T. denticola*. In conclusion, the in-house real-time PCR method developed in our laboratory can provide information about relative amount of five bacterial species present in subgingival plaque in patients with periodontitis. It is likely that such a test could be used in dental diagnostics in assessing the efficacy of any treatment to reduce the bacterial burden.

## 1. Introduction

Periodontal disease is caused by the loss of supporting tissues within the periodontium (gingivae, periodontal ligament) and a degeneration of alveolar bone support. Chronic periodontitis (CP) is the most prevalent type of periodontal disease, an inflammatory disease induced by pathogenic bacteria in the subgingival plaque [1]. CP is a complex disease and has a multifactorial ethiopathogenesis in which the dental plaque (biofilm) is the primary initiator of the inflammatory process [2,3,4]. Gradually, plaque components adjacent to the soft tissues of the supporting periodontium have an effect, leading to gingival inflammation and, in severe cases, alveolar bone necrosis and tooth exfoliation [5,6].

Numerous changes in the oral microbiota, such as pH and nutrition, may lead to the development of human periodontitis. Maintaining a good balance of the oral microbiome has a significant impact on oral health and plays an important role in the physiological, nutritional, and immune development of individuals [6].

Although there are more than 700 bacterial species that colonize the soft tissues of the mouth or develop a biofilm on teeth, only a small percentage of these bacteria have the potential to destroy periodontal tissues [7,8]. According to current scientific knowledge, five periodontopathic bacteria are associated with transformations in the physiological oral habitat and include *Aggregatibacter actinomycetemcomitans*, *Porphyromonas gingivalis*, *Tannerella forsythia* (previously *Bacteroides forsythus*), *Prevotella intermedia*, and *Treponema denticola* [4,5,7,9,10].

Due to genetic diversity, certain strains of *A. actinomycetemcomitans* are associated with periodontitis. Serotypes a and b correlate with severe periodontitis, whereas serotype c is commensal in healthy individuals [4]. *P. gingivalis* has been detected in more than 85% of dental plaque in chronic periodontitis patients [9]. *P. gingivalis* is one of the main causative factors in periodontitis pathogenesis [10]. The presence of *T. forsythia* and *T. denticola* in dental biofilm has been significantly associated with severe forms of periodontitis, whereas *P. intermedia* shows moderate pathogenicity [11,12,13,14]. These bacteria are Gram-negative and require anaerobic growth conditions, except for *A. actinomycetemcomitans*, which is facultative anaerobic [4,14].

When these Gram-negative bacteria prevail in the oral cavity, plaque forms, which leads to inflammation and the progression of the disease [15,16]. Also, the plaque is permeated by circulatory channels that allow nutrients to flow in and waste products or metabolites to flow out. The accumulation of a metabolite beyond a threshold level induces changes in gene expression within the community. This communication mechanism that biofilm bacteria use is called quorum sensing. Additionally, based on similar characteristics and/or symbiotic relations, certain bacterial species will only congregate with certain other bacterial species. The association between *P. gingivalis* and *T. denticola* is an illustration of such symbiosis. *P. gingivalis* uses succinate, which *T. denticola* produces from amino acid fermentation in subgingival plaque. Conversely, *P. gingivalis* produces certain fatty acids that can stimulate *T. denticola* growth [17].

Considering the number of bacterial species colonizing the surface of the oral mucosa and their significant role in oral health and disease, accurate identification is crucial. Traditional techniques have been based on culture and microscopy, biochemical tests, immunofluorescence staining, and antibiotic sensitivity. These methods are labor-intensive and costly, providing sometimes inconsistent results [15,17]. Recently, molecular DNA-based techniques became a standard for the identification of bacteria directly from clinical samples. Several methods have been developed and became commercially available for the quick and precise identification of periodontopathogenic bacteria. These assays are based on several molecular technologies, including extraction of the bacterial DNA from the plaque samples and amplification of the specific DNA sequences of the target periodontal pathogen. Polymerase chain reaction has become a standard diagnostic tool in dentistry. Real-time PCR, due to its specificity and sensitivity, is a rapid and efficient method to detect, identify, and differentiate microorganisms [15,16]. It is valuable tool for the detection of bacteria that are difficult to grow in culture.

The real-time PCR method and DNA-strip technology (DNA–DNA hybridization) are commercially available assays in microbiological laboratory procedures. The company Hain Lifescience GmbH, (Nehren, Germany) introduced the Micro-IDent^®^ test identifying five periodontopathogenic bacteria (*A. actinomycetemcomitans*, *P. gingivalis*, *T. forsythia*, *P. intermedia*, *T. denticola*) [4,5]. Multiplex PCR is followed by a reverse hybridization where amplified PCR products bind to 16S ribosomal RNA gene probes corresponding to specific bacteria [16].

The purpose of this study was to develop an in-house real-time PCR method as a diagnostic tool for the detection of periodontal pathogens in patients with periodontitis. To confirm the validity, i.e., the advantages and limitations of this method, the obtained results were compared with the results obtained with the commercially available micro-IDent^®^ diagnostic test.

## 2. Results

The comparative results regarding the detection of five periodontopathogenic bacteria with both diagnostic methods are shown in Table 1. All samples detected as negative with micro-IDent^®^ test were confirmed with a real-time PCR method that showed a specificity of 100% for all bacteria. Measure of agreement between two methods for the presence of all bacteria indicated by kappa statistics was very high. The best agreement was found in the case of *T. forsythia* 0.924 and the lowest was for *T. denticola* 0.819. 

One positive sample of *A. actinomycetemcomitans* with micro-IDent^®^ test was negative with real-time PCR. This finding results in a sensitivity of 97.5% of the real-time PCR considering micro-IDent^®^ test results as the reference standard. In the case of *A. actinomycetemcomitans*, the measure of agreement was 0.918.

In the case of the *P. gingivalis*, only one sample showed a discrepancy. The micro-IDent^®^ test detected that sample as positive while the real-time PCR was found to be negative. This discrepancy showed a sensitivity of 96.88%. Measure of agreement kappa was 0.862.

The biggest discrepancy was found with the *P. intermedia*. Two samples, positive by the micro-IDent^®^ test, were negative by real-time PCR and a comparison of the results for *P. intermedia* showed a sensitivity 95.24% and the measure of agreement kappa was 0.846.

The best agreement between the micro-IDent^®^ test and real-time PCR method was for *T. forsythia* and *T. denticola*. All samples that were positive by testing with the micro-IDent^®^ test were fully confirmed with the real-time PCR method, which ultimately resulted in a sensitivity of 100% for both bacteria.

Real-time PCR results were presented as logarithmic values, based on the Ct distribution in four groups that correspond to micro-IDent^®^ categorization: negative (no bacteria detected), one plus (+; <10^4^), two (++; 10^4^–10^6^), and three (+++; >10^6^).

Serial dilutions of the extracted DNA of known concentrations were used to determine analytical sensitivity, as shown in Figure 1. After constructing the diagram of the standard curves, a regression analysis was made, and from the data of the standard deviation intercept ((S.D.) intercept) and the slope of the curve, the detection limit (*LoD*) and the quantification limit *(LoQ*) were determined. They were determined using equations:$$LoD=\left(3.3\times \left(\frac{\sigma }{s}\right)\right);\;LoQ=(10\times \left(\frac{\sigma }{s}\right))\;;\;\sigma \text{—}S.D.\;\mathrm{of}\;\mathrm{intercept}\;\mathrm{and}s\text{—}\mathrm{slope}$$

The limit of quantification was 100 genome copies, while the limit of detection was less than 10 genome copies.

The agreement of the results obtained with two different measurements, for each bacterium separately, is shown in Figure 2.

The results are shown for each score individually, and for each bacteria there is an agreement of the results for the negative score. This means that every negative result (marked 0 on the *x*-axis and a blue column) made by the micro-IDent^®^ test was also confirmed by the real-time PCR method.

Furthermore, the results of score 1 obtained by the micro-IDent^®^ test for *A. actinomycetemcomitans* and *P. gingivalis* bacteria by the real-time PCR method gave one negative result each (score 0), i.e., two negative results for *P. intermedia* bacteria, that is, they represented a false negative score for real-time PCR.

Regarding the discrepancy in scores (1–3) for the bacterium *A. actinomycetemcomitans*, the results differed in all scores (1–3) for only one sample per score that was analyzed by real-time PCR and classified as a lower score. In the positive samples of *P. gingivalis*, there is a difference between the results of the real-time PCR test compared to the results of the micro-IDent^®^ in five samples. One sample analyzed by real-time PCR in score 1 showed one score lower than the result obtained by micro-IDent^®^, while in scores 2 and 3, two samples in each score analyzed by real-time PCR showed a score lower than to the result obtained by micro-IDent^®^. In the case of positive samples of *P. intermedia* bacteria, the difference was also in five samples. The results differed for two samples each in scores 1 and 3, while the difference in score 2 was observed only in one sample. The results obtained by real-time PCR showed one score lower.

Positive results for the bacterium *T. forsythia* differed in scores 1 and 2 for only one sample in each score. Real-time PCR results show one score higher than those obtained by micro-IDent^®^. In the case of *T. denticola* bacteria, the biggest difference is in score 2, where five samples analyzed by real-time PCR show a higher score compared to the results obtained, and in score 1, one sample shows a higher score. In the case of bacteria *T. forsythia* and *T. denticola* in score 3, there was no difference in the results obtained by both methods.

Spearman’s correlation factor was calculated for all results shown in Table 2 from which the correlation factors of the two measurements for all bacteria were extremely high, which means that the correlation was strong.

## 3. Discussion

In this study, we compared real-time PCR test developed in our laboratory with the established micro-IDent^®^ diagnostic test to validate the efficacy of our real-time PCR method for identifying high-risk periodontal bacteria present in subgingival plaque. Consequently, we included in the analysis five periodontopathogenic bacteria that are most commonly associated with the onset of periodontal disease: *Aggregatibacter actinomycetemcomitans*, *Porphyromonas gingivalis*, *Tannerella forsythia*, *Prevotella intermedia*, and *Treponema denticola* [4,5,7,9,10].

The results of our study conducted on subgingival samples using two different methods—the in-house method developed in our laboratory and the utilization of a commercial test—were compared, revealing an exceptionally strong correlation. These findings indicate a high analytical sensitivity and specificity of the developed in-house method. All samples identified as negative by the micro-IDent^®^ test were confirmed negative by the real-time PCR method, resulting in a specificity of 100%. Such high specificity was observed for all analyzed bacteria.

Although this real-time PCR method cannot be applied for the precise quantification of pathogens, qualitative analysis based on cycle threshold (Ct) values provides valuable information regarding detection, i.e., whether the analyzed periodontopathogenic bacteria are present in subgingival plaque. Additionally, we had the ability to use a range of Ct values and expressed the results semi-quantitatively. As outlined in the Materials and Methods section, we categorized the results based on Ct values into four distinct scores. This approach contributed to the method’s utility in determining the range of bacteria present in the sample. A zero-score denoted negative samples, while the remaining three scores represented positive samples. Subsequently, we compared positive results obtained by real-time PCR with those obtained by the micro-IDent^®^ test.

In the case of *T. forsythia* and *T. denticola*, no false negative results were observed, thus, yielding a sensitivity of 100%. However, for *A. actinomycetemcomitans*, *P. gingivalis*, and *P. intermedia*, the sensitivity was slightly lower, as the results obtained by real-time PCR were negative, whereas those same samples tested positive with the micro-IDent^®^ test, i.e., it registered in score 1. It is noteworthy that the detection threshold using the PCR method appeared to be higher compared to the micro-IDent^®^ test, resulting in no false positive results from the PCR method.

Furthermore, among the positive samples, slight discrepancies were observed regarding categorization into scores. For the bacterium *A. actinomycetemcomitans*, the results differed in all scores (1–3) for only one sample per score analyzed by real-time PCR, which were classified into lower scores. In the case of the bacterium *P. gingivalis*, among the positive samples with scores 1–3, the real-time PCR test results varied from the micro-IDent^®^ results in five samples. One sample analyzed by real-time PCR in score 1 had a lower value than that obtained by micro-IDent^®^, while in scores 2 and 3, two samples each analyzed by real-time PCR had lower values than those obtained by micro-IDent^®^. For the bacterium *P. intermedia*, the test results differed for two samples in scores 1 and 3, while the difference in score 2 was observed for only one sample.

For the bacteria *T. forsythia* and *T. denticola*, the difference in scores was evident by a shift of one score higher for the results obtained by the PCR method. In *T. forsythia*, the difference was observed in two samples, one each in scores 1 and 2. For the bacterium *T. denticola*, one sample from score 1 showed score 2 with the PCR test, while the greatest difference was noted in the results of score 2, where the PCR test indicated score 3 for five samples. It is worth noting that for bacteria *T. forsythia* and *T. denticola*, there was no difference in results for score 3. We emphasize that these are borderline values, which slightly lowers the sensitivity of our method, as demonstrated in Table 1.

Regardless of the observed differences, the results analyzed by the kappa statistical test (measure of agreement of positive scores) between the two methods for the presence of all indicated bacteria were very high. The best agreement was found for T. forsythia at 0.924, while the lowest was for T. denticola at 0.819.

Comparing the results of previously published studies with those of our study is often difficult due to the different bacteria analyzed, the variable sampling processes used to collect subgingival plaque for examination, and the various methodologies used to identify and quantify those bacteria [4,18,19,20,21]. Therefore, we compared the results partially.

In their study, Storm et al. managed to increase specificity from 18.5% to 92% by adjusting the PCR detection threshold. [22]. Comparing such adjustments, we are willing to conclude that the detection threshold in our method is well-adjusted, with a specificity of 100%. The relationship between quantitative results obtained by real-time PCR and the micro-IDent^®^ test for all five bacterial species is presented in Table 2.

Eick et al. compared the results obtained using the real-time PCR method and the micro-IDent^®^ test. These methods were used to analyze subgingival samples collected from 25 patients with periodontitis. The correlation coefficients of the two methods were between 0.62 (*T. denticola*) and 0.74 (*P. gingivalis*). In our study, all correlations between the two methods were highly significant; the correlation coefficients were between 0.917 (*T. denticola*) and 0,976 (*A. actinomycetemcomitans*). This is higher than in the mentioned study that reached a coefficient of 0.74 when they compared micro-IDent^®^ test and real-time PCR, which we attribute to our specific selection of primer sequences [21].

Molecular tests, such as real-time PCR, have become key diagnostic tools for bacterial infections due to their higher sensitivity and shorter processing time compared to conventional methods such as culture. Considering that real-time PCR can detect even low copy numbers of microorganisms, numerous PCR assays have been developed, but their performance characteristics vary due to the use of primers targeting different gene regions, different PCR reagents, and instruments for analysis [23,24,25,26].

Various commercial assays are currently available for identifying periodontal pathogenic bacteria, including micro-IDent^®^ (Hain Lifescience GmbH, Nehren, Germany) [4]. However, the advantages of a real-time PCR method include high sensitivity and specificity, as well as the capability for rapid and cost-effective qualitative and quantitative detection of previously mentioned oral pathogens. This novel molecular approach can be utilized in clinical microbiology for diagnosing oral diseases.

The results of this study have confirmed the validity of our in-house real-time PCR method for the identification and semi-quantification of high-risk periodontal bacteria present in subgingival plaque.

### Study Limitations

The limitation of this study is that this real-time PCR test can provide only information about the relative amount and type of five bacterial species present in subgingival plaque in patients with periodontitis. It is likely that such a test could be used in dental diagnostics in assessing the efficacy of any treatment to reduce the bacterial burden.

## 4. Materials and Methods

### 4.1. Samples

A total of 50 Caucasian patients, 21 male and 29 female, average age 45–50 years with clinical symptoms of localized periodontitis were included in the study. Periodontitis was diagnosed if the patient had a pocket probing depth ≥4 mm. All of them had shown signs of traumatic occlusal forces and needed oral rehabilitation. Dental examination and microbiological sampling were performed by a dentist, who at the time of the examination determined the degree of inflammation in the oral cavity.

### 4.2. Sample Collection and Storage

The sampling area was isolated with cotton rolls. The tooth surface was cleaned with 70% ethanol and dried with sterile cotton swabs. Samples were taken from the deepest pockets of the diseased areas using five sterile paper points, which were inserted into the gingival crevice for 15 s and then placed in a sterile 1.5 mL tube for molecular testing. The sample was transported to a molecular laboratory, having a stability of 7 days at temperatures of 2–8 °C.

### 4.3. DNA Extraction

In tube with periodontal sample, 500 µL of saline was added, mixed vigorously for 10 s to wash the bacterial cells from the paper points, and then the DNA was extracted using NucleoSpin^®^-Microbial kit (Macherey-Nagel, Düren, Germany) according to the manufacturer’s recommendations. The quantity of extracted DNA was determined spectrophotometrically at 260 nm using a NanoDrop ND-1000 Spectrophotometer (Thermo Fisher Scientific Inc., Waltham, MA, USA), and its purity by the ratio of the absorbance measured at 260 and 280 nm. Afterward, the DNA was split into aliquots for real-time PCR and micro-IDent^®^ test.

### 4.4. Micro-IDent^®^ Test

The micro-IDent^®^ test (Hain Lifescience, Nehren, Germany) includes diagnostics of the five most common periodontopathogenic bacteria belonging to high pathogenic potential: *Aggregatibacter actinomycetemcomitans*, *Porphyromonas gingivalis*, *Prevotella intermedia*, *Tannerella forsythia*, and *Treponema denticola* This molecular test is based on PCR and DNA STRIP technology. The test consists of two parts: multiplex amplification with biotinylated primers and a reverse hybridization.

#### 4.4.1. Amplification

All reagents needed for amplification were included in the amplification mix A (primer–nucleotide mix) and amplification mix B (PCR buffer, MgCl_2,_ and Taq polymerase) and were optimized for this test. The samples were amplified in a 50 µL reaction volume consisting of 5.0 µL of template DNA and 45 µL of reaction mixture (10 µL amplification mix A and 35 µL amplification mix B). Negative control was included in the test, consisting of 5.0 µL of sterile water added to 45 µL of the reaction mixture. PCR cycling was performed in SimpliAmp Thermal Cycler (Thermo Fisher Scientific Inc., Waltham, MA, USA). Cycling conditions included an initial denaturation step at 95 °C for 5 min, 10 cycles of 95 °C for 30 s and 58 °C for 2 min, 20 cycles of 95 °C for 25 s, 53 °C for 40 s and 70 °C for 40 s, and a final extension step at 70 °C for 8 min.

#### 4.4.2. Hybridization

The biotinylated amplicons were denatured and incubated at 45 °C with hybridization buffer and strips coated with two control lines (amplification control and hybridization control) and five species-specific probes. PCR products bound to their respective complementary probe and highly specific washing step removed any nonspecific bound DNA. Streptavidin-conjugated alkaline phosphatase was added, the samples were cleaned, and hybridizations products were visualized by adding a substrate for alkaline phosphatase. The strips were paste on the analysis forms and the results were read using the evaluation sheet. The range of the white background of the bands and the conjugate control was set to 100%, and the value of each measured band was set relative to the resulting percent staining relative to the control. A clear band indicated a positive result. The presence of an invisible-looking band indicated a negative result. The results are reported in four categories: negative (no bacteria detected), one plus (+; <10^4^), two (++; 10^4^–10^6)^, and three (+++; >10^6^). According to the manufacturer, the cut-off of the test is set to 10^3^–10^4^ genome equivalents [21].

### 4.5. Real-Time PCR

Real-time PCR was carried out in a reaction volume of 50 µL consisting of 5.0 µL of the isolated DNA as template and 25 µL Power Syber Green PCR Master Mix (Thermo Fisher Inc., Waltham, MA, USA), 18 µL sterile water, and 2 µL (20 µM) of bacteria specific primer pair. Several primer pairs already published in respective papers were tested for optimal conditions and efficiency of amplification. The best results have been obtained by primers shown in Table 3 [19]. Primer sequences belong to 16S ribosomal RNA gene. The primer concentrations were the same for all assays. The positive controls were made by pooling genomic DNA of the five positive targeted bacteria previously confirmed by both micro- IDent^®^ and real-time PCR test analysis. The 5.0 µL of positive and 5.0 µL negative control were included in each analysis run. Negative controls consisted of ultrapure water. The PCR was carried out in ABI Prism 7500 Real-Time PCR System (Applied Biosystems, Waltham, MA, USA). All amplifications and detections were carried out in a MicroAmp optical 96-well reaction plate. The cycling conditions were initial denaturation at 95 °C for 10 min, followed by 40 cycles of denaturation at 95 °C for 5 s and annealing at 60 °C for 34 s each. The accumulation of PCR products was observed at each cycle by monitoring the increase in fluorescence of the reporter dye from dsDNA binding SYBR Green. 

Furthermore, after the PCR the specificity of the amplification was assayed with the use of melting curves, which was constructed in the range of 60 °C to 95 °C. 

For each individual bacterium, from plaque samples, the concentration of DNA was determined by the quantitative commercial test PeriodontScreen Real-TM PCR (Sacace Biotechnologies, Como, Italy) according to the manufacturer’s recommendations.

From the resulting concentrations for each bacterium, serial 10-fold dilutions were made from 10^5^ to 10^1^ dilutions to determine the quantitative range of real-time PCR.

Standard curves for each individual bacterium were constructed from the obtained results.

### 4.6. Reporting the Results

The real-time PCR results are reported as value of threshold cycle (Ct), which is defined as the first cycle in which fluorescence is detectable above the baseline and is inversely proportional to the logarithm of the initial number of template molecules. Thresholds are usually set within the exponential phase of PCR. If target is present in the reaction, amplification signal will emerge from the baseline. The intersection of the threshold and amplification plot produce a Ct value. Ct values can be used for qualitative and semi-quantitative assays under condition that thresholds must be fixed to maintain consistency in Ct value calculations each time the assay is running [27]. 

The DNA from plaque samples obtained from 50 patients with clinical symptoms of localized periodontitis were analyzed by two methods: micro-IDent^®^ test and real-time PCR. Micro-IDent^®^ results were categorized as follows: 0: no band: negative; +: a weak band: weak positive; ++: a clearly visible band: positive; +++: a clearly visible band: intense positive. The same samples were analyzed by real-time PCR and the average range of Ct values corresponding to the micro-IDent^®^ results was determined separately for each positive value (+; ++; +++) and negative value (0). The Ct range of real-time PCR analysis for each of five bacteria was calculated as follow: *Aggregatibacter actinomycetemcomitans*, *Porphyromonas gingivalis*, *Prevotella intermedia*: 0 (negative) > 22.1; (+) 18.1–22; (++) 14.1–18; (+++) < 14; *Treponema denticola* and *Tannerella forsythia*: 0 (negative) > 26.1; (+) 22.1–26; (++) 18.1–22; (+++) < 18. Real-time PCR results were presented as logarithmic values, based on the Ct distribution in four groups that corresponds to micro-Ident® categorization: negative (no bacteria detected), one plus (+; <10^4^), two (++; 10^4^–10^6^), and three (+++; >10^6^).

### 4.7. Statistical Analysis

The sensitivity was determined as the number of samples found positive both by micro-IDent^®^ test and by real-time PCR divided by the numbers of positive results by the micro-IDent^®^ test. The specificity was calculated as the number of negative results found concomitant by both tests divided by the numbers of negative results by the micro-IDent^®^ test. The degree of agreement between two methods real-time PCR was assessed by the kappa statistic. Correlation was determined by using Spearman test. Analyses were performed with statistical Package Software for Social Science, version 26 (SPSS Inc., Chicago, IL, USA).

## 5. Conclusions

We have developed a simple, efficient, and reliable test for detecting periodontopathogenic bacteria. According to statistical analyses, our in-house method demonstrates the highest agreement with the micro-IDent^®^ test. Considering the shorter analysis time and lower reagent costs associated with our in-house method compared to micro-IDent^®^, it presents potential applicability in routine clinical practice.

## Figures and Tables

**Figure 1 ijms-25-05097-f001:**
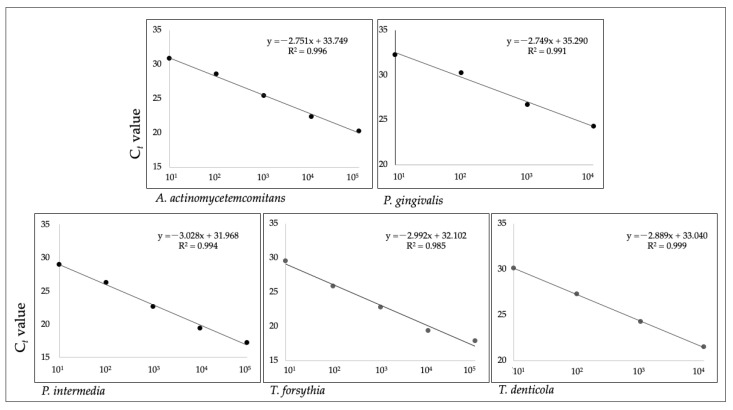
The standard curves of the quantitative real-time PCR. Serial dilutions of extracted DNA from *A. actinomycetemcomitans*, *P. gingivalis*, *P. intermedia*, *T. forsythia*, and *T. denticola* and number of genome copies shown on the *x*-axis. The *y*-axis shows Ct value.

**Figure 2 ijms-25-05097-f002:**
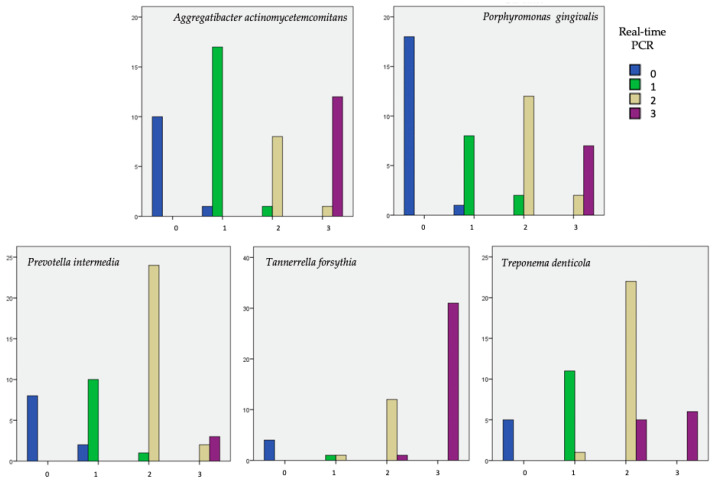
Comparison of the results obtained with two different methods: micro-IDent^®^ test, shown on the *x*-axis (results classified into 4 groups: 0: negative; 1, 2, and 3: positive) and real-time PCR marked according to the colors shown in the legend. The *y*-axis shows the number of respondents in each category.

**Table 1 ijms-25-05097-t001:** Comparison of detection of five periodontopathogenic bacteria in 50 plaque samples by micro-IDent^®^ and real-time PCR.

	Detection by Real-Time PCR (Bacteria/Plaque Sample)					
	**Score**					
		**0**	**<10^4^**	**10^4^** **–10^6^**	**>10^6^**	**Total**
**Detection by micro-IDent^®^**	Score					
*A. actinomycetemcomitans*	0	10	0	0	0	10
	+	1	17	0	0	18
	++	0	1	8	0	9
Sensitivity 97.5%; specificity 100%	+++	0	0	1	12	13
Measure of agreement kappa 0.918	Total	11	18	9	12	50
*P. gingivalis*	0	18	0	0	0	18
	+	1	8	0	0	9
	++	0	2	12	0	14
Sensitivity 96.88%; specificity 100%	+++	0	0	2	7	9
Measure of agreement kappa 0.862	Total	19	10	14	7	50
*P. intermedia*	0	8	0	0	0	8
	+	2	10	0	0	12
	++	0	1	24	0	25
Sensitivity 95.24%; specificity 100%	+++	0	0	2	3	5
Measure of agreement kappa 0.846	Total	10	11	26	3	50
*T. forsythia*	0	4	0	0	0	4
	+	0	1	1	0	2
	++	0	0	12	1	13
Sensitivity 100%; specificity 100%	+++	0	0	0	31	31
Measure of agreement kappa 0.924	Total	4	1	13	32	50
*T. denticola*	0	5	0	0	0	5
	+	0	11	1	0	12
	++	0	0	22	5	27
Sensitivity 100%; specificity 100%	+++	0	0	0	6	6
Measure of agreement kappa 0.819	Total	5	11	23	11	50

Micro-IDent^®.^ results were categorized as follows: 0: no band: negative; +: a weak band: weak positive; ++: a clearly visible band: positive; +++: a clearly visible band: intense positive.

**Table 2 ijms-25-05097-t002:** Spearman’s correlation factor of the two different measurements: micro-IDent^®^ test and real-time PCR for the five bacterial species.

		Micro-IDent^®^ Test				
**Real-Time PCRRT-PCR**		** *Aa.* **	** *Pg.* **	** *Pi.* **	** *Tf.* **	** *Td.* **
*Aa.*	Correl.Coeff.	0.976 **				
*Pg.*			0.967 **			
*Pi.*				0.949 **		
*Tf.*					0.966 **	
*Td.*						0.917 **

*Aa.* = *A. actinomycetemcomitans*, *Pg.* = *P. gingivalis*, *Pi.* = *P. intermedia*, *Tf.* = *T. forsythia*, *Td.* = *T. denticola*. ** Correlation is significant at the 0.01 level.

**Table 3 ijms-25-05097-t003:** Target bacteria and their species-specific primers used in real-time PCR.

Bacterial Species	Strain	Primer Sequences (5’-3’)	Product Size (bp)
*Aggregatibacter actinomycetemcomitans*	ATCC 43718	TGTGCCTTAGGGAGCTTTGAGACA GCAACAAAGGATAAGGGTTGCGCT	106
*Porphyromonas gingivalis*	ATCC 33277	CGGGATTGAAATGTAGATGATGATGG ACACCTTCCTCACGCCTTACG	200
*Prevotella intermedia*	ATCC 25611	TGGAACTGAGACACGGTCCAAACT TTGGCTGGTTCAGACTTACGTCCA	102
*Treponema denticola*	ATCC 35405	AGAGCAAGCTCTCCCTTACCGT TAAGGGCGGCTTGAAATAATGA	108
*Tannerella forsythia*	UB22	ACACCTCCTTTCTGGAGCAGTCTT AACCGAGACATCCCAGCTTCCTTT	138

## Data Availability

The data presented in this study are available on request from the corresponding author.

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
