# Peer review of "Real-Time PCR Method as Diagnostic Tool for Detection of Periodontal Pathogens in Patients with Periodontitis"

_ijms, 2024, doi:10.3390/ijms25105097_

Round 1

Reviewer 1 Report

Comments and Suggestions for Authors

The reviewed manuscript describes an in-house protocol for PCR-based detection of five bacterial species associated with chronic periodontitis. The topic itself is interesting and timely, as oral microbiota has gained increasing attention from scientists. However, there are several serious issues in the study that make it unsuitable for publication in its present form.

Major issues:

  1. The authors are requested to underline the limitations of currently existing PCR-based methods. Many such tests have been reported recently, necessitating clarification of the contribution of the presented study to the field.
  2. The methodology of the study seems to be lacking. There were no positive controls made either from genomic DNA of the targeted bacteria, plasmids with cloned amplicons, or gBlocks. Therefore, the analytical sensitivity and specificity of the primers were not assessed, which is necessary for a proper test.
  3. Quantification of bacteria in samples using Cq values only, without determining the limits of detection and quantification, with unknown PCR efficacy, as well as without a standard curve, appears excessively straightforward. This approach seems doubtful because real PCR efficiency can vary greatly, affecting the obtained results.

Minor issues:

  1. The RT-PCR acronym can be somewhat confusing because it also means PCR coupled with reverse transcription. In that sense, it would be better to use another acronym, e.g., qPCR (quantitative PCR).

Author Response

The authors wish to thank reviewer for valuable critical input, and hope that the changes introduced in the paper will now meet their acceptance criteria.

Below we described the changes. The changes are highlighted in red. Through track changes it is visible in the text.

Reviewer 2 Report

Comments and Suggestions for Authors

The aim of this study was the development of a real-time PCR test as a diagnostic tool for the detection and differentiation of five periodontopathogenic bacteria: Aggregatibacter actinomycetemcomitans, Porphyromonas gingivalis, Tannerella forsythia, Prevotella intermedia and Treponema denticola in patients with periodontitis. The sensitivity for A. actinomycetemcomitans was 97.5%, for P. gingivalis 96.88%, and for P. intermedia 95.24%. The sensitivity for T. forsythensis and T. denticola was 100%.

Please ensure T. forsythia spelling is standardised through manuscript (not T. forsythensis).

Diagnosis should be based on 2018 EWP/AAP classification. For example, line 261: should be localised periodontitis; not local chronic periodontitis. There is also insufficient information on patient profile – where were they recruited from, patient characteristics.

Do explain where there was not a negative control.

Do also explain why these five ‘periopathogenic’ bacteria are crucial for identification, in light of current understanding of the oral dysbiosis, and other techniques such as 16S rRNA sequencing and high throughput sequencing that provide a far more comprehensive picture of the microbiome.

Comments on the Quality of English Language

Minor editing required.

Author Response

(The authors gave the same response as above.)

Reviewer 3 Report

Comments and Suggestions for Authors

The manuscript describes the development of a real-time PCR as a diagnostic tool for the detection and differentiation of five periodontopathogenic bacteria, and the comparison with the semiquantitative commercially available test based on the PCR hybridization method. The contents of the manuscript seem suitable for the publication in the International Journal of Molecular Sciences. The specific points are as follows;

[Suggestions]
Introduction (L. 47-49);
and Discussion (L. 170-172):

"Although there are more than 700 bacterial species that colonize the soft tissues of the mouth or develop a biofilm on teeth, only a small percentage of these bacteria have the potential to destroy periodontal tissues [7, 8]."

"Although the physiological flora of the oral cavity is composed of many different types of bacteria, only a few are responsible for development of severe and progressive periodontal diseases [7, 8]."

Are the authors really sure? Most of the dental researchers in the world do not simply think so; Numerous environmental changes in the oral microbiota, such as pH, anaerobiosis and nutrition, may lead to an accumulation of periodontitis-associated bacteria in the subgingival sulcus, resulting in the initiation of human periodontitis. Both qualitative and quantitative changes in subgingival plaque biofilms in periodontal pockets are thought to be highly associated with both the initiation and progression of periodontitis.

Author Response

(The authors gave the same response as above.)

Round 2

Reviewer 1 Report

Comments and Suggestions for Authors

Many thanks to the authors for their detailed and careful replies to comments in the review. However, the manuscript still needs substantial corrections before possible publication.

1.    The authors have acknowledged the limitations of their study, which is indeed a valid point. However, the initial question about issues with the currently used PCR tests remains unanswered. These issues need to be clearly stated in the manuscript for readers to better understand the problem. Without this clarification, the relevance of the study is compromised, as many similar tests have already been reported and commercialized.

2.    The question about analytical characteristics suggests additional experiments to determine sensitivity and specificity. In this regard, DNA samples with known concentrations can be used to establish the Limit of Detection (LoD) and Limit of Quantification (LoQ) of the analysis. Without these procedures, application of the reported test could lead to false-negative or false-positive results, and the manuscript cannot be accepted.

Author Response

The authors wish to thank reviewer for valuable critical input, and hope that the changes introduced in the paper will now meet their acceptance criteria.

Reviewer 2 Report

Comments and Suggestions for Authors

I thank the authors for the clarifications to my comments and the manuscript is now in publishable form. 

Author Response

(The authors gave the same response as above.)

Round 3

Reviewer 1 Report

Comments and Suggestions for Authors

Many thanks to the authors for their thoughtful replies to comments in the second review. The quality of the manuscript has substantially improved because of the additional experiment defining analytical characteristics of the designed qPCR. However, the description of this experiment seems to lack important details, such as how were samples for DNA purification obtained, DNA purification procedure, DNA quantification and dilution, and how were LoD and LoQ defined. Also, the acronym “RT-PCR” still presents in Figure 2.

Author Response

(The authors gave the same response as above.)
